# Integration of Intercultural Learning into an International Advanced Pharmacy Practice Experience in London, England

**DOI:** 10.3390/pharmacy9010037

**Published:** 2021-02-11

**Authors:** Ellen Schellhase, Ishmum Hasan, Stephanie Hendricks, Monica L. Miller

**Affiliations:** College of Pharmacy, Purdue University, West Lafayette, IN 47907, USA; elschell@purdue.edu (E.S.); hasan14@purdue.edu (I.H.); hendri50@purdue.edu (S.H.)

**Keywords:** experiential learning, intercultural learning, pharmacy education

## Abstract

As the number of international advanced pharmacy practice experiences (APPEs) continues to grow, this is an opportunity to incorporate intercultural learning (ICL) to further advance student pharmacist training. Purdue University student pharmacists participated in a clinical research focused APPE in London, England. To prepare for this APPE, students completed a one-credit course focused on intercultural learning and travel preparation. The purpose of this report is to describe the implementation and assessment of ICL during this course and international APPE. The course includes interactive ICL activities, reflective assignments, and personalized assessments. During the eight-week APPE, student pharmacists worked on an individualized Intercultural Development Plan^®^, which includes ICL activities, focused reflection, and check-ins. ICL was assessed using the Intercultural Development Inventory^®^ (IDI^®^) at the beginning of the course and at least four weeks after APPE completion. Student APPE feedback was also reviewed for evidence of ICL. Twenty-seven students completed the course and APPE from 2018 to 2020. The average IDI developmental orientation (DO) before the course was 91.7, placing students in minimization. The average perceived orientation was 120.9, placing students in acceptance. There were 18 students who completed the post-APPE IDI: 12 students demonstrated growth in the DO (range: 1.5–23.72), and six students experienced a decrease in their DO. Intercultural learning can be implemented and assessed as part of an international APPE.

## 1. Introduction

According to the World Health Organization (WHO), transitions of care and the third Global Patient Safety Challenge, one of the key steps for ensuring medication safety, is communication and patient engagement. Effective communication between patients and healthcare providers ensures accurate and complete medication information transfer at interfaces of care [1,2]. However, a key challenge defined by WHO is the cultural, organizational, and professional barriers to collaboration, communication, and integration between healthcare professional groups, healthcare professionals, and patients. Not only can communication create barriers to receiving optimal care, but it has been found that there are also sociocultural challenges that add to this issue within healthcare [3]. As a result of the inability to communicate efficiently and effectively, it can be difficult to demonstrate compassionate care to those with different cultural backgrounds [3]. 

In order to bridge the gap between healthcare professionals and patients, there is an increasing demand to enhance intercultural competence within health education programs, particularly within pharmacy [4,5,6]. There are many different solutions for integrating cultural awareness education into curricula [7]. One way is through experiential training programs that offer learners international and global health-focused experiences. These experiences provide unique learning opportunities that allow learners the chance to engage with different health systems, increase medical knowledge, provide care to underserved populations, practice and enhance their cultural intelligence, and improve their abilities to empathize [8]. They also provide a focused opportunity for increasing cultural awareness or cultural intelligence through intercultural learning. Due to the focus on developing global health competencies, which include cultural awareness, there is a high number of pharmacy programs (~79%) that offer some type of introductory and/or advanced pharmacy practice experience in global health and/or offer medical mission trips (~65%) [9,10]. 

One of the main objectives of global health training is to foster intercultural learning (ICL). Self-awareness, awareness of others in context to self, managing emotions and thoughts, and bridging cultural gaps/adapting behavior in effective and appropriate ways are thought to be the four core skill areas of ICL [11,12,13]. While this is a life-long process of development, there are various ways to track individual progression. Tracking of progress remains challenging given the variety of definitions surrounding culture and competence [14]. There is, however, agreement that intercultural competence is based on one’s use of effective and appropriate knowledge, skills, and attitudes [15,16]. Recently published research utilized the Consortium of Universities for Global Health (CUGH) competency statements to assess knowledge, skills, and attitudes of student pharmacists who participated in an international clinical experience. The authors described a significant improvement in several ICL areas, including interprofessional value and communication skills, appreciation of cultural differences, and global health perspectives, compared to students who did not participate in an international advanced pharmacy practice experience (APPE) [17]. 

Although there is literature describing health professional students obtaining intercultural learning experience from courses, study abroad programs, and international experiential training, there is limited data documenting intercultural growth objectively before and after an international experience [18,19,20,21]. This case report describes an intercultural learning curricular design including an assessment plan for student pharmacists at Purdue University, who participated in an international APPE at St. Bartholomew’s Hospital (Barts) in London, England. 

## 2. Design and Assessment

### 2.1. Intercultural Learning Curricular Design

The Purdue University College of Pharmacy has found success in partnering a pre-requisite preparatory course with an international APPE to aid in students’ longitudinal development of global health competences, in particular ICL [22]. The purpose of the course affiliated with the London-based APPE is to help prepare student pharmacists for active and effective engagement and collaborative research during their international APPE. To meet this purpose, 50% of the course is focused on knowledge-based activities to prepare the students for working in a foreign health system. Topics include: the National Health Service (NHS), disease state management within NHS, site protocols, and guidelines, along with travel tips. The other 50% of the course time is dedicated to ICL. Topics covered include cultural dimensions, communication and conflict styles, and intercultural core competencies (Table 1). The incorporation of ICL activities and experiences was designed to address Bloom’s affective domain, while the other more traditional learning objectives address the cognitive and psychomotor domains of learning [23,24]. When developing the ICL activities used in this course, course faculty worked with specialists from the Purdue Center for Intercultural Learning, Mentorship, Assessment, and Research, and consulted the Intercultural Learning Hub (HubICL). This online resource houses ICL activities that have been used by ICL specialists in a variety of settings [25].

During the APPE, student pharmacists spend eight weeks actively engaging with Barts preceptors in patient care and clinical research in either cardiology, nuclear pharmacy, or oncology. There are a few unique aspects of this APPE curriculum. The first is the requirement to produce an abstract that can be submitted to a professional meeting by the completion of the APPE. Secondly, students travel together in groups of three or four and share a living space in London. The last aspect is that the APPE includes working through an individualized Intercultural Development Plan^®^ (IDP^®^), which includes ICL activities and focused reflection. The IDP is a tool generated after completion of the Intercultural Development Inventory^®^ [26]. In addition to the individualized plan, students can participate in check-ins with course faculty to discuss their progress and share their intercultural journey. 

### 2.2. Assessment Plan

Due to the interconnected ICL curriculum between the pre-requisite course and the APPE, students completed the IDI at the beginning of the course and at least four weeks after APPE completion (Figure 1) [27]. To identify examples of changes in knowledge, skills, and attitudes, student course feedback and survey responses were reviewed by the authors. 

#### Intercultural Development Inventory^®^ (IDI) Assessment Tool

The Intercultural Development Inventory^®^ (IDI) is a validated 50-question online assessment that explores intercultural understanding, cultural intelligence, global effectiveness, and cross-cultural adaptation at an individual level [28,29]. The assessment takes approximately 15–20 minutes to complete, and currently costs $12 for students use. The IDI provides quantitative data on the Perceived Orientation (PO), Developmental Orientation (DO), and Orientation Gap (OG). The PO score on the IDI represents where one places themself along the Intercultural Development Continuum^®^ (IDC^®^) or how one sees themselves when interacting with culturally diverse individuals and groups. The DO score indicates the primary orientation toward cultural differences and commonalities as assessed by the IDI. The DO is the perspective most likely used in situations where cultural differences and commonalities need to be bridged [29]. Lastly, the OG score is the difference between PO and DO scores, which can be used to provide insight into one’s self-awareness. The IDI is beneficial when working with individuals seeking to improve and grow interculturally. It helps identify an intercultural baseline, identify gaps, and create an individualized Intercultural Development Plan^®^ (IDP). The IDP is focused on the student’s individual DO, and provides suggested activities and reflective prompts to move along the IDC. The IDI is one of the few intercultural assessments that evaluates not only the perceived skill set, but also the actual ability to interact with cultural difference. Furthermore, this tool is presented as an Intercultural Development Continuum^®^, which can help the individual understand where their current skill level is, as well as inform them of their next stage of growth (Table A1 and Figure A1) [16,28,29]. 

This case report received approval from the Purdue Human Research Protection Program and Institutional Review Board (IRB) via Cayuse IRB, study # IRB-2020-1656.

## 3. Outcomes

This case report assessed 27 students who completed the pre-requisite course and the London APPE from 2018 to 2020. Out of the 27 students, 18 (67%) completed the post-London APPE IDI. Table 2 outlines the IDI results for the participants. The average IDI DO before the course was 91.7, placing most students in Polarization or Minimization. The average PO before the course was 120.9, placing students in Acceptance. A paired *t*-test was performed, which demonstrated the overall DO change to be statistically significant (*p* = 0.01) and a non-statistically significant change in the PO (*p* = 0.07). After ICL in the course and the London APPE, 12 students demonstrated growth in both DO and PO; four students had an increase in their orientation. The average DO growth was 13.5 points (1.5–23.72). There were six students with a decrease in their DO; one student moved to the previous orientation. The average decrease in DO was 6.9 points (0.39–12.58). Overall, there was a decrease in OG by an average of five points. Figure 2 outlines the overall orientations of the group before and after the APPE. To support the intercultural growth identified in the IDI, Table 3 contains student comments that are examples of intercultural competence knowledge, skills, and attitudes that were learned.

## 4. Lessons Learned

This case demonstrates the value of adding intentional, individualized, and intercultural learning to an APPE, even in a setting often perceived by some to be minimally culturally different [30]. Students, in this case, were in a new healthcare practice setting, living and working with classmates, and in many cases experiencing their first time abroad. Most student pharmacists experienced growth in their ICL based on the IDI score changes observed. Additional growth was observed in the qualitative feedback from the APPE, which further supports the intercultural learning that students experienced during their APPE. The intercultural knowledge, skills, and attitudes were learned and applied in the professional setting. This type of layered learning achieves the recommendations for cultural competence recommended in the Accreditation Council for Pharmacy Education Accreditation (ACPE) Standards and the Center for Advancement of Pharmacy Education (CAPE) Educational Outcomes, and can be applied in many different environments, whether they are domestic or international [5,6].

This case also demonstrates the utility of the IDI as an assessment tool for quantitatively illustrating intercultural learning during international APPEs. While there are many examples of its utility in study abroad, there is a lack of examples of its use in pharmacy experiential learning [7]. There are other intercultural assessment tools described in the literature; however, most do not focus on their applicability in evaluating growth in APPEs [18]. This tool was particularly selected for several key reasons. The first was its ability to identify both the perceived and objective skill level for working across cultural difference [29]. Another advantage is the generation of the IDP. The IDP allows students to craft their own ICL learning plan based on their specific skill level/IDI orientation. It also allows students the opportunity to take ownership of their own ICL growth instead of following a prescribed learning plan that may or may not meet their goals or needs. Lastly, as course faculty are not with students during APPEs, this tool allows for a guided learning plan that can be used outside of the classroom. For our learners, most started with DOs in denial, polarization, or low minimization. The most growth in ICL came from students who were lower on the continuum in the beginning. Continuing to provide ICL guidance through the IDP extends the opportunity for student learning during the APPE.

While most students did experience ICL growth as represented by the IDI scores, there were some who experienced an IDI regression. Study abroad research using the IDI has highlighted reasons for regression and theories as to why most students do not progress beyond Minimization from study abroad participation [31]. One known reason for regression is the boomerang effect. For students who travel abroad, this can be a time of idealized life and idealized responsibilities/freedoms. When they return to their home culture, they are faced with their “real” life, and this re-entry can be jarring for students, which in some cases is reflected in a regression of their ICL [31]. With the instructional assessment plan used, students may have experienced the boomerang effect already four weeks into their return. Of the six students who experienced a regression, three were in minimization, and remained there despite decreasing their IDI score. This regression could be potentially mitigated using pre-return debriefs to provide focused time to think about returning to the US and maintaining ICL growth. 

Another possible reason for regression is unmet expectations during the international experience, or a sense of being totally overwhelmed by the cultural differences that students and, as a result, seeking out only what is familiar to them [32]. Living and working abroad is known to be stressful, and this experience may be too much change at once, causing the student to become overwhelmed. When they reach the overwhelmed phase, without proper reflection and debriefing, this can derail a student’s intercultural learning. Additionally, students can have unspoken expectations for their study abroad, and when these expectations are not met, it can impact their overall learning, and they can associate a negative experience to learning within a new culture.

### 4.1. Limitations

The number of students assessed in this case report was limited. The small sample size and limited statistical analysis also limit the applicability of the results. Additional students are necessary to determine the significance of a longitudinal, individualized ICL learning plan on student growth. The IDI is both developmental and grounded in theory, and research has demonstrated the value of the IDI even when non-probability samples are used; this is a common limitation of researching study abroad, and a limitation of this case [31]. Research using the IDI has indicated that IDI growth or regression is considered meaningful when there is more than a seven-point change in the IDI [16]. If this level of significance was applied to the student pharmacists in this case report, five students (29.4% of the cohort) would report meaningful growth, and only three students (17.6%) would report meaningful regression. 

Studies have also shown that the more facilitated the intervention, the more meaningful growth can be measured [33]. While the ICL activities in the pre-requisite course were well-facilitated, the activities during the APPE were less structured. The activities in the course were associated with a grade, which can motivate students to complete it. The IDI assessment and IDP completion as part of the APPE were not associated with a grade. This variation in accountability could impact intercultural growth. A larger cohort would be required to determine the impact of facilitation for this curricular design. 

### 4.2. Future Plans and Replication Opportunities

Future plans include refining the ICL activities and adding additional assessments. The HubICL has several additional activities that can be added to the course, and includes activities targeted as specific intercultural competencies [24]. The findings of this case report will also guide the creation of a practice-focused training manual for this practice site, which will assist future students with their intercultural learning, and serve as a guide that can be tailored to other Purdue international APPEs. To further enhance the ICL that student pharmacists are working on during the international APPE, the faculty would like to integrate additional guided debriefs. Self-reflection, which is a significant part of the IDP, could also be evaluated to inform about student learning. As COVID-19 has forced courses to adapt to virtual and hybrid offerings, faculty have also considered using an online certificate course to provide structured intercultural training [34]. Authors have also begun to assess the value of using other intercultural assessment tools. Students in the 2019–2020 cohort took the Cultural Intelligence (CQ) assessment as a baseline assessment [35]. The preliminary data indicated that the student self-assessment was moderate (in the middle of world-wide norms) across all four capabilities assessed: drive, knowledge, action, and strategy. Future plans include adding a post-APPE CQ assessment and analyzing correlations between the IDI and CQ for this student population. In addition to the IDI and CQ, there are other validated ICL assessment tools that may meet the needs of a program, measure a particular construct of ICL, and can be paired with assessments of qualitative feedback within an APPE [24]. This case report can serve as a model for integrating and assessing intercultural learning within an APPE, allowing readers to adapt and change the content and/or assessment methods to work within their own APPE setting.

As more institutions consider the integration of ICL into international APPEs, many of the activities and assessments in this case report can be easily replicated. One limitation to using the IDI as an assessment for ICL is that it requires administration and debriefing from an IDI qualified administrator. Replication is also possible beyond international APPEs. Focused ICL can be integrated into other experiential learning experiences, such as introductory pharmacy practice experiences and domestic APPEs. While there are examples of ICL implementation in didactic curriculum, there is a lack of data within experiential curricula. Implementing ICL within experiential learning can help institutions empower their students while meeting ACPE standards and CAPE outcomes. 

## Figures and Tables

**Figure 1 pharmacy-09-00037-f001:**
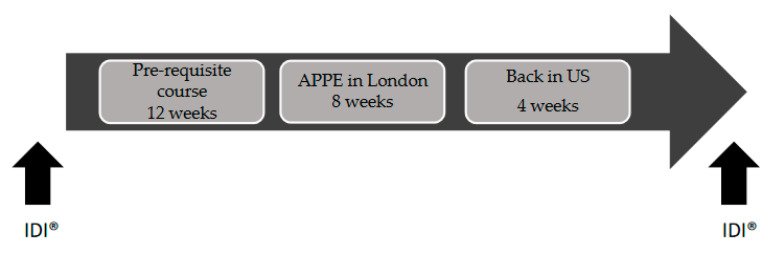
Instructional assessment plan. Before departing to London for their international APPE rotation, students complete a 12-week pre-requisite course. The students took the IDI prior to the pre-requisite course and again four weeks after returning to United States.

**Figure 2 pharmacy-09-00037-f002:**
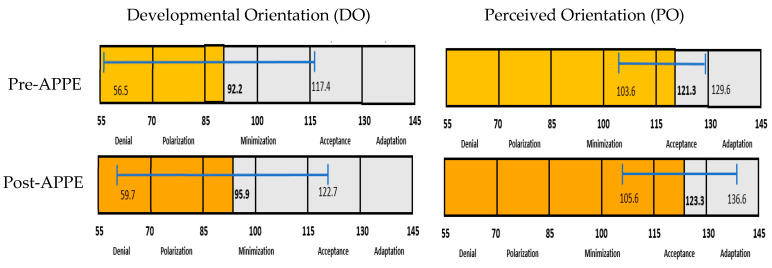
Changes in Developmental Orientation (DO) and Perceived Orientation (PO) scores. Detailed overview of the IDI orientation differences from before and after the London APPE. The figure shows the average, minimum, and maximum points of student IDI scores. Overall, DO, and PO scores improved after the London APPE rotation.

**Table 1 pharmacy-09-00037-t001:** Examples of intercultural learning activities in the course.

Activity/Course Content	Description
Cultural Introduction	Students introduce themselves culturally by sharing several of their cultural identities, a quote that has meaning to them, and a goal they have for the course related to their projected growth.
Conflict Style Inventory	Students are introduced to the four primary conflict styles, including corresponding actions. They subsequently are asked to assess their own styles and share these amongst the class. Students then discuss scenarios in which they may experience conflict during their international APPE.
Cultural Dimensions	Students learn about the different cultural dimensions and how these are different for cultures around the world. To contextualize these, students work through case scenarios of how these could be expressed within the United Kingdom (UK). They are then asked to compare and contrast their own personal preferences around the dimensions with those of the global UK population. With the comparisons, they are asked to brainstorm how they will work across the possible differences.
Emotional Hot Buttons	Students participate in an activity that highlights different emotional hot buttons (i.e., being interrupted, soft or loud talking, lack of eye contact). They identify their three biggest hot buttons and discuss these in pairs. The class debriefs the most common hot buttons and their connection to the cultural dimensions. Students also discuss how to approach common hot buttons during their international APPE.
Emotional Regulation Moments	This pre-requisite course provides students an opportunity to begin regulating their emotions through the use of in-class meditation. Each class begins with a guided meditation.
Pacing Communication Styles	Students are introduced to the different pacing communication styles. They participate in a role play activity where they act out a conversation with people from the three different styles. During the conversation, they must utilize a prescribed pacing style, and then the class debriefs on this experience.

**Table 2 pharmacy-09-00037-t002:** IDI Results.

Pre-Requisite Course Year	Average Pre-Course Perceived Orientation (PO)	Average Pre-Course Developmental Orientation (DO)	Average Post-APPE PO	Average Post-APPE DO
2018–2019 (*n* = 15)	120.3	91.7	121.9 (*n* = 11)	93.2 (*n* = 11)
2019–2020 (*n* = 12)	121.5	91.6	125.45 (*n* = 7)	100.11 (*n* = 7)
Total (*n* = 27)	120.9	91.7	123.2 (*n* = 18)	95.9 (*n* = 18)

**Table 3 pharmacy-09-00037-t003:** Examples of student reflections demonstrating intercultural learning.

		Definition	Reflection
Knowledge	Healthcare System	Students described similarities and/or differences of the healthcare system within the context of insurance/coverage schemes or policies/regulations that influence healthcare.	“This rotation opened my eyes to alternative ways of doing things both within a medical/pharmacy setting and without it. I was able to more fully grasp how a single-payer health system works and see its advantages and disadvantages, which allowed me to better understand the obstacles facing America’s current healthcare flux situation.” “I learned how profit focused American healthcare is and how little we focus on the actual character of healthcare professionals—care for the patients.”
Skills	Communication—Cross Cultural	Students described the ability to communicate within a cultural context.	“Communication was the most difficult aspect of the rotation. I thought since we both speak English that communication would be easy. How I was wrong. The cultural differences between the different styles of communication completely caught me off guard. Compared to their counterparts in the US, UK pharmacists and healthcare practitioners were addressed differently, and they tend to be very polite and formal in even short messages.”
Attitudes	Self-Awareness	Students described increased conscious knowledge of their own culture, feelings, and motives.	“I was able to work on my self-reflection skills and becoming more self-driven. The students were not given much instruction but were expected to ask for work, so I had to overcome my fear of annoying my preceptors/being in the way by asking for more work. This was a cultural difference I had to adjust to while on this rotation, but it allowed me to become more assertive.”
	Appreciation—Culture	Students described an appreciation for London’s culture.	“I learned a great deal about the differences in the culture in London and other countries. I grew up in a small town in Indiana, so I have not had a great deal of exposure to diversity beyond my experiences at Purdue.” “I liked how the UK promotes healthy diet to their citizens. I had to adjust my lifestyle a little bit but also raised my awareness to what I eat and how much daily exercise I need.”
	Appreciation—Experience	Students described their experience as amazing, invaluable, and life-changing.	“The experience was unlike any other rotation that I have had. During my rotation, there were also pharmacists from other countries such as France and Portugal, who were also rotating in the hospital. I have also learned about the practice of pharmacy in their respective regions—it was more than just interesting.”

## Data Availability

Data sharing is not available. No new data were created or analyzed in this study. Data sharing is not applicable to this article.

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
