# Peer review of "Integration of Intercultural Learning into an International Advanced Pharmacy Practice Experience in London, England"

_pharmacy, 2021, doi:10.3390/pharmacy9010037_

Round 1

Reviewer 1 Report

This is an interesting and important article. I am glad to see attention being paid to intercultural learning.

An important caution for this case report is about generalization. Although the authors did not use this term, statements such as “The findings of this case report will also guide the creation of a practice-focused training manual, which will assist future students with their intercultural learning and serve as a guide which can be tailored to the international APPE” (lines 240-243) cause concern to me as a reader. The intercultural experiences for these students was from an English based country (USA) to another English based country (England). One might question the intercultural differences, although they exist, they could not parallel differences between the US and Asia or Africa, for example. So inferences made from this case report have to be contextualized and explicitly discussed, either in the limitations or in the replication opportunities.

I appreciated Table 1 on page 3 as it gives the reader an idea of the expectations and activities in in the program. Line 87 says Bloom’s affective domain was used to help design the course. I would like to know more about how Bloom’s affective taxonomy was used, as it is not evident from the main body of the text and it is not in the reference list. If it is not referenced, the then this should be deleted from the paper.

Descriptive statistics were used for the analysis. Without inferential statistics, it is difficult to ascertain whether there was actual improvement. With 27 participants, although only 18 completed the posttest, at least a t-test could have been used. Results and conclusions should be treated with caution.

Page 7 includes a discussion about possible reasons for the regression of some of the students. Again, caution rears its head. These are conjectures from the authors with no evidence to support the possible reasons. One simple reason could be that some students simply were not affected by the intercultural program, but this is not suggested.

Overall, it is an interesting paper, but validity lies in the inferences we make from studies, and I have some worries with this paper. The authors need to state that this is a case, but due to small numbers, the parallels between the two cultures, and the descriptive statistics one should not generalize to other situations. Instead, recommendations for more research in this area is what is needed.

Reviewer 2 Report

See attached file. 

Reviewer 3 Report

This is a nicely written and original paper describing a 12-week didactic preparatory course followed by an 8-week research/clinical APPE in London.   Authors did an excellent job explaining their research objective, their methods, and thoroughly described and evaluated their results.  Researchers' approach is reproducible and can be utilized by others planning an international APPE to capture student growth in the intercultural learning domain.  The article was easy to read and follow, tables and figures were clear and helped understand manuscript content.

Author Response

The authors appreciate the feedback from this reviewer.  

Round 2

Reviewer 2 Report

N/A